# Running from Stress: Neurobiological Mechanisms of Exercise-Induced Stress Resilience

**DOI:** 10.3390/ijms232113348

**Published:** 2022-11-01

**Authors:** Marta Nowacka-Chmielewska, Konstancja Grabowska, Mateusz Grabowski, Patrick Meybohm, Malgorzata Burek, Andrzej Małecki

**Affiliations:** 1Laboratory of Molecular Biology, Institute of Physiotherapy and Health Sciences, Academy of Physical Education, 40-065 Katowice, Poland; 2Department for Experimental Medicine, Faculty of Medical Sciences in Katowice, Medical University of Silesia, 40-752 Katowice, Poland; 3Department of Anaesthesiology, Intensive Care, Emergency and Pain Medicine, University Hospital Würzburg, 97080 Würzburg, Germany

**Keywords:** stress, stress resilience, anxiety, depression, neuropsychiatric disorders, physical activity, exercise

## Abstract

Chronic stress, even stress of a moderate intensity related to daily life, is widely acknowledged to be a predisposing or precipitating factor in neuropsychiatric diseases. There is a clear relationship between disturbances induced by stressful stimuli, especially long-lasting stimuli, and cognitive deficits in rodent models of affective disorders. Regular physical activity has a positive effect on the central nervous system (CNS) functions, contributes to an improvement in mood and of cognitive abilities (including memory and learning), and is correlated with an increase in the expression of the neurotrophic factors and markers of synaptic plasticity as well as a reduction in the inflammatory factors. Studies published so far show that the energy challenge caused by physical exercise can affect the CNS by improving cellular bioenergetics, stimulating the processes responsible for the removal of damaged organelles and molecules, and attenuating inflammation processes. Regular physical activity brings another important benefit: increased stress robustness. The evidence from animal studies is that a sedentary lifestyle is associated with stress vulnerability, whereas a physically active lifestyle is associated with stress resilience. Here, we have performed a comprehensive PubMed Search Strategy for accomplishing an exhaustive literature review. In this review, we discuss the findings from experimental studies on the molecular and neurobiological mechanisms underlying the impact of exercise on brain resilience. A thorough understanding of the mechanisms underlying the neuroprotective potential of preconditioning exercise and of the role of exercise in stress resilience, among other things, may open further options for prevention and therapy in the treatment of CNS diseases.

## 1. Introduction

Stress is a well-characterized factor causing changes in the brain function and structure by, for example, modifying the neurochemical and neuroendocrine processes, and causing physiological alterations [1]. Long-term stress impairs the function of negative feedback loops within the hypothalamic–pituitary–adrenal (HPA) axis so that they no longer serve their purpose. It may cause the increased secretion of both the corticotropin-releasing hormone (CRH) and glucocorticosteroids (GCs) [2]. Cortisol/corticosterone (CORT), acting via steroid receptors, inhibits the production and secretion of hormones in brain structures [3]. In particular, glucocorticoid receptors (GR) and mineralocorticoid receptors (MR) are expressed at high levels in the hippocampus, amygdala, prefrontal cortex, and other limbic and midbrain structures, where they modulate the neural circuitry and neuroendocrine systems that underlie the behavioural responses to stress [4]. The stress-induced synthesis of GCs [5] causes the sustained hyperactivity of neurons, leading to excitotoxicity and the generation of reactive oxygen species (ROS) [6,7]. A long-term exposure to social stress affects the levels of monoamine in the limbic regions, which are known to be particularly sensitive to stress [4]. Following stress, decreased levels of dopamine (DA) in the medial prefrontal cortex (mPFC), increased norepinephrine (NE) and serotonin (5-HT) in the dentate gyrus, and decreased NE in the dorsal raphe, have been observed [8,9]. In addition, stress exposure disrupts neuronal plasticity as a result of suppressed neurogenesis, cell atrophy, or enhanced apoptosis [10]. In humans, chronic stress is recognized as a major risk factor for several psychiatric disorders, including anxiety and depression [11,12]. Interestingly, most individuals do not develop such disorders after experiencing stressful life events and are thus thought to be resilient. In this context, stress resilience is defined as an active, adaptive process, which allows the individual to cope successfully with acute stress, trauma, or chronic adversity [13]. As with humans, in laboratory animals there is a clear relationship between the disturbances induced by stressful stimuli, especially long-lasting stimuli, and cognitive deficits [14]. In numerous studies, it has been demonstrated that chronic stress leads to the development of depression- or anxiety-like behaviours [15,16,17,18]. However, the remaining stressed animals, those considered “resilient”, exhibit some physiological symptoms without deleterious behavioural changes. Resilient animals do not exhibit social avoidance, anhedonia-like symptoms (for example reduced sucrose consumption), or metabolic disturbances defined by over-eating, a preference for high-fat food, obesity, and obesity-associated changes (e.g., central leptin resistance) [19]. It should be noted that stress-resistant individuals are characterized by a high sympathetic system activity, a low parasympathetic system activity (for example, high heart rates), as well as a low production of glucocorticoids in response to stress [20,21], a high resistance to mental illness [22], and a high motor activity (active exploratory reactions, aggression, etc.). Additionally, mice with a high baseline physical activity (high voluntary activity in “running wheels”) demonstrated a resistance to chronic social stress compared to animals neglecting exercises [23,24]. 

Both preclinical and clinical studies have suggested that moderate-intensity exercise is beneficial during a lifetime because it promotes a variety of physiological adaptations that reduce the anxiety response [25,26]. Chronic and regular physical activity has a positive effect on the central nervous system (CNS) functions, contributes to the improvement of one’s mood and cognitive abilities (including memory and learning) [27,28], and correlates with an increase in the expression of neurotrophic factors and markers of synaptic plasticity [29], as well as a reduction in the inflammatory factors [30]. Studies published so far show that the energy challenge caused by physical exercise can affect the CNS by improving cellular bioenergetics, stimulating the processes responsible for the removal of damaged organelles and molecules, and attenuating the inflammation processes reviewed in [31]. In addition to the well-documented neural mechanism for exercise, the facilitation of synaptogenesis and stabilization of newly formed synapses were found in the cortical region after chronic treadmill exercise [32,33]. Finally, regular physical activity brings another important benefit: increased stress robustness. The evidence from human and animal studies suggests that a sedentary lifestyle is associated with stress vulnerability, whereas a physically active lifestyle is associated with stress robustness [34,35], which is defined as a state of both stress resistance and stress resilience.

Revealing the mechanisms involved in the beneficial effect of physical activity/exercise on stress resilience could contribute to an understanding of how it can induce positive physiological improvements, protect against the effects of stressful events, and prevent or minimize several CNS diseases. In this context, this review will consider the following question: what molecular mechanisms/signalling pathways related to physical activity can contribute to stress resilience, and can physical exercise contribute to the resilience of the brain to stress in some neurological disorders? Here, we discuss the findings only from animal experiments and studies of the molecular and neurobiological mechanisms underlying the preventive effect of exercise on stress. Even though exercise might act as a stressor, it has been widely demonstrated that it reduces the deleterious effects of harmful stressors. For clarification, we focus on the neuroprotective potential of exercise as a preconditioning to a further exposure to a variety of stressors, considering the molecular targets involved in exercise-induced neuroprotection: increased neurogenesis, the expression of neurotrophic factors, the modulation of neurotransmission and HPA activity, the reduction in cerebral inflammation and oxidative stress, and improved cognition [36,37].

## 2. Literature Search Strategy

A comprehensive PubMed search was conducted to identify studies investigating the role of exercise in stress resilience and the possible molecular mechanisms underlying exercise-induced stress resilience. The literature search was conducted between July and August 2022. The results were sorted by date and included all papers from 2010 to 2022. For further information, older publications were also included. Original (experimental and clinical) studies, letters, and reviews, written in English, were included.

Studies that did not address physical activity/exercise for stress resilience were excluded. The following terminology was applied: (physical activity AND stress resilience), (“exercise” OR “physical activity”) AND (“stress-resistant” OR “stress-resilience” OR “brain resilience”) AND (“psychiatric disorders” OR “depression” OR “anxiety” OR “brain diseases”), (“exercise prior to stress” AND “brain”). While reviewing the literature, special attention was paid to the frequently interchangeable terms of exercise and physical activity. It is worth noting that physical activity should be defined as any bodily movement produced by the skeletal muscles which results in an energy expenditure, while exercising is a physical activity that is planned, structured, repetitive, and purposeful.

## 3. Results

A PubMed database search revealed 935 potentially relevant articles; 252 studies concerned physical activity or exercise and stress resilience. Most records were excluded because they did not address physical activity or exercise or the stress resilience outcome. Of the 252 articles, 156 studies referred to clinical research, 68 studies addressed pre-clinical research, and 28 were reviews. Of the 68 studies, conducted on mice or rats, 36 were not eligible, resulting in 30 articles that involved exercise prior to stress and focused on the brain molecular targets of stress resilience (Figure 1, flow chart).

## 4. Discussion

As a result of the high prevalence of stress-related psychiatric disorders, including major depression, post-traumatic stress disorder (PTSD), and anxiety, and the limited efficiency of current pharmacological treatments, there is a need to identify novel therapeutic strategies. It has been suggested that mechanisms involving a decrease in neurogenesis, changes in the neurotransmitter levels, the modulation of neuroinflammation, brain plasticity, and glutamate imbalance contribute to the stress-related disorders reviewed in [38]. All of these aspects are affected by exercise. Physical exercise has been proposed as an effective and low-cost therapy to prevent CNS disorders, especially those related to long-lasting stress. Previous studies have mainly focused on the therapeutic effects of exercise on stress-induced CNS disorders, while only a few have described exercise as a preconditioning for a subsequent exposure to stress. Recently, exercise has been associated with cognitive improvement and stress resilience [13,39,40,41], but the underlying neurobiology remains unclear. In this review, attention is focused on animal models, since these have played a crucial role in searching for and identifying the neural mechanisms underlying exercise-induced stress resilience. Table 1, Table 2, Table 3, Table 4, Table 5 and Table 6 sets out a summary of animal studies regarding the stress resilience induced by physical exercise and studies utilizing the exercise paradigm (voluntary wheel running, treadmill, or swimming exercise) before stress. For more details please see Appendix A.

### 4.1. Neurogenesis, Neuroplasticity, and Cognition 

The hippocampus is a very well-studied brain structure because of its structural plasticity and vulnerability to a wide range of experiences [42,43,44]. It is well known that exogenous corticosterone or stress induces dendritic atrophy in the apical dendrites of the CA3 regions and has a minor effect on the CA1 region and the dentate gyrus of the hippocampus [45]. Stress or hypercortisolism suppresses the hippocampal excitability and long-term potentiation, which is always associated with an impaired hippocampus-dependent memory [46]. Changes in neuroplasticity following social adversities are strongly related to the individual vulnerability to stressful stimuli [9]. Repeated exercise improves the cognitive functions, spatial learning and memory, and also reverses stress-induced changes in the behavioural parameters reviewed in [47]. The beneficial impact of exercise is probably regulated by hippocampal neurogenesis. Moreover, physical exercise is one of the strongest positive physiological modulators of the hippocampal structure and function [48]. Therefore, it is not surprising that both forced (treadmill) and non-forced spontaneous (activity wheel) physical exercise increased hippocampal neurogenesis and cell proliferation [49,50]. For example, voluntary exercise increased adult hippocampal neurogenesis in the rostral hippocampus of eight-month-old female mice and also stimulated neurogenesis in adult mice subjected to early life stress, affecting the postmitotic DCX-positive cells [51]. Finally, exercise minimized, or even reversed, the stress-induced changes in hippocampal neurogenesis, which correlated with improved cognitive functions in aged mice [52] and humans [53]. Treadmill exercise following or at the same time as an exposure to the social stressors induced by social defeat [54] or a PTSD model [55,56] resulted in the prevention of anxiety- and depression-like behaviour. Here, it is noteworthy that, as a result of a review by Holmes (2014) [57], the anxiolytic effects of exercise in rodents are most consistent when the subjects have previously been exposed to a stressor, thus distinguishing the preventive and the therapeutic effects of exercise. 

In a recent study by Sun et al. (2021) [58], swimming exercise reduced the vulnerability of mice to CUMS (chronic unpredictable mild stress) and contributed to the AKT/GSK-3β/CRMP2 pathway and microtubule dynamics (assessed by measurement of Tyr-tubulin expression in the hippocampus) mediated protective effects on neuroplasticity. Five weeks of swimming exercise also prevented mice from engaging in the depression-like behaviour induced by CUMS. The authors concluded that exercise contributes to stress resilience via the AKT/GSK-3β/CRMP2 pathway and by maintaining microtubule dynamics [58]. On the other hand, six weeks of endurance exercise provided stress resilience to the morphological changes (dendritic length, dendrite spine density) induced by a later exposure to PSS (predator scent stress). Animals who had exercised before stress showed an increase in resilience of the cytoarchitecture of the neurons in the dentate gyrus subregion of the hippocampus [59]. Another possible mechanism for the neuroprotective effects of exercise prior to stress in regard to the hippocampal region was shown by Kim et al. (2011). These authors showed a protective effect of eight weeks of preconditioning exercise on hippocampal neuronal death induced by an exposure to repeated restraint stress and the i.c.v. injection of kainic acid. Prior chronic treadmill running suppressed kainic acid-induced hippocampal neuronal death, assessed by the number of Nissl-positive neuronal cells in the hippocampal CA3 region of the stressed mice. These results were accompanied by lower ROS levels (lower malondialdehyde (MDA) and nitrite levels), and enhanced levels of pCREB (cAMP response element-binding protein), which were mediated by ERK1/2 and CaMKII [60]. Among the molecules regulated by the neuronal cells to cope with stressful stimuli, CREB is known to play a crucial role in neuronal survival [61]. It could be suggested that the activation of pro-survival pathways is involved in the phenomena of stress resilience (Table 1). Taken together, the data presented above support the hypothesis that chronic and regular exercise has neuroprotective properties in regard to neurodegenerative disorders.

**Table 1 ijms-23-13348-t001:** Summary of animal research focusing on neurogenesis, neuroplasticity, and cognition.

No.	Strain	Sex	Age	Physical Activity	Stress Model	Main Outcome	Reference
1	C57BL/6J	male	5 weeks	swimming	unpredictable mild stress	modulation of AKT/GSK-3β/CRMP2 pathway; maintaining microtubule dynamics	[58]
2	C57BL/6J	male	N/A	treadmill	restraint stress	attenuated hippocampal cell death; CREB activation; decreased ROS levels	[60]
3	Long–Evans	female	7 weeks	voluntary wheel running	traumatic stress	ameliorated stress-induced maladaptive behaviours	[57]
4	Sprague–Dawley	male	2.5 months	treadmill	predator scent stress	improved morphological features of neurons	[59]

AKT—AKT kinase; CREB—cAMP response element-binding protein; GSK-3β—glycogen synthase kinase-3β; ROS—reactive oxygen species.

### 4.2. Neurotransmitter Systems

One of the proposed mechanisms for the different responses to stress among individuals may be related to differences in the baseline physical activity. The involvement of the dopamine (DA) system has been proposed as one of the possible mechanisms. Growing evidence suggests that stress causes changes in the brain areas associated with a reward, and the dysfunction of mesolimbic DA neurons has a negative impact on the behavioural responses to social stress [62,63]. Voluntary wheel running as a natural reward for rodents may be associated with the DA system [64,65]. 

In a recent study by Zhang et al. (2021), the authors divided wild-type mice into those with a low and those with a high baseline physical activity, based on the results from three days of voluntary wheel running. The mice with a high baseline physical activity displayed a greater resilience to chronic social defeat stress (CSDS), showing a higher expression of tyrosine hydroxylase (TH) and more TH-positive neurons in the ventral tegmental area (VTA) than the mice with a low baseline physical activity. Furthermore, the activation of the TH neurons in the VTA of mice with a low physical activity by the DREADD method increased their physical activity and resilience to stress. The opposite effects, namely a reduced level of wheel running and increased susceptibility to stress, were observed after the inhibition of the TH neurons in the VTA of mice with a high physical activity. The authors concluded that the dopamine system may affect the resilience to CSDS, possibly via an alteration of the baseline physical activity [23] (Table 2). Similarly, rats with a low adaptive response to treadmill exercise showed structural anomalies in their noradrenergic locus coeruleus (LC). Furthermore, rats with a high adaptation to exercise showed higher levels of serum ACTH, increased pERK activation in response to two hours of physical restraint stress, and diminished processing of fear-associated memories compared to the rats selectively bred for a low adaptation [24]. In a study by Mul et al. (2018) that used a molecular and viral gene transfer approach, it was demonstrated that the sustained inhibition of ΔFosB in the NAc (nucleus accumbens) of mice is likely to be a contributor to greater stress resilience. The authors demonstrated that 42 days of voluntary wheel running induced ΔFosB in the subregions of the NAc and ameliorated stress-induced social avoidance behaviour and anhedonia [66]. The chronic induction of ΔFosB in the NAc, which is a crucial reward-related brain area, is strongly associated with neuronal adaptations to drugs of abuse and stress [67]. In previous studies, wheel running for several weeks has been shown to result in an increase in ΔFosB in the NAc in both rats [61] and mice [68]. Rats housed in an enriched environment with running wheels recovered much faster from the inescapable footshock (IFS) paradigm, which is a model of PTSD, than animals housed under standard conditions. Moreover, this behavioural recovery was correlated with an increased cell proliferation in the hippocampus, a decrease in the tissue levels of noradrenaline, and the increased turnover of 5-HT in the prefrontal cortex and hippocampus [69]. In female rats, six weeks of voluntary wheel running before uncontrollable tail-shock stress buffered against the behavioural changes—avoidance and exaggerated fear—produced by stress [70]. Tanner et al. (2019) have suggested a role for the dorsal raphe nucleus (DRN) in mechanisms underlying the observed stress resilience induced by wheel running in female rats. This suggestion is based on the results from studies by Greenwood et al. (2003) in which wheel running exercise constrained the activation of DRN 5-HT neurons during stress in male rats [71]. Additionally, six weeks of wheel running lessens the stress-induced 5-HT neural activity, assessed by c-Fos protein expression in the DRN and lateral ventral region of the bed nucleus of the stria terminalis (BNST), which probably contributes to the exercise-induced prevention of the behavioural consequences of uncontrollable stress [72]. Recently, in a mice model of chronic stress, the GABA receptors within the habenula and a subregion of the mPFC were shown to play an important role in stress resilience and vulnerability [73,74]. Another possible mechanism underlying stress resilience involves GABAergic plasticity in the hippocampus. Chowdhury et al. (2019), using electrophysiological techniques, showed that exercise promotes resilience to food-restriction stress by enhancing the GABAergic inhibition of pyramidal neurons in the dorsal hippocampus of female rats [75]. 

**Table 2 ijms-23-13348-t002:** Summary of animal research focusing on neurotransmitter systems.

No.	Strain	Sex	Age	Physical Activity	Stress Model	Main Outcome	Reference
1	C57BL/6J	male	7–8 weeks	voluntary wheel running	chronic social defeat stress	increased TH expression and number of TH-positive neurons in the VTA	[23]
2	C57BL/6	male	10 weeks	voluntary wheel running	chronic social defeat stress	induced ΔFosB in the subregions of the NAc	[66]
3	Sprague–Dawley	female	N/A	voluntary wheel running	uncontrollable tail shock	alleviated behavioural consequences of stress	[70]
4	Fischer F344	male	7–8 weeks	voluntary wheel running	uncontrollable tail shock	attenuated the dorsal raphe nucleus 5-HT activity	[71]
5	Sprague–Dawley	female	4 weeks	voluntary wheel running	food-restriction stress	enhanced the GABAergic inhibition of pyramidal neurons in the dorsal hippocampus	[75]

5-HT—serotonin; NAc—nucleus accumbens; TH—tyrosine hydroxylase; VTA—ventral tegmental area.

### 4.3. Neurotrophic Factors

The neurotrophic factors are a group of proteins possessing similar structures that are synthesized in peripheral tissues innervated by sensory and sympathetic neurons and in the neurons of some brain structures [76]. Neurotrophins diminish neuronal degeneration and induce and enhance synaptic plasticity [77]. Among the members of the neurotrophin family, the brain-derived neurotrophic factor (BDNF) is one of the most intensively studied. BDNF expression is sensitive to stress. Findings from previous studies have shown that both acute and chronic stress decrease the BDNF mRNA in the hippocampus. Recent findings suggest that the neural plasticity supported by the trophic mechanisms is vital for establishing and maintaining stress resilience [78,79]. However, behaviourally resilient rodents do not necessarily show a higher basal BDNF expression [80,81,82]. 

Many studies have shown the magnitude of the exercise-induced expression of trophic factors in the brain and periphery, such as the BDNF, insulin-like growth factor (IGF-1), vascular endothelial growth factor (VEGF), and the inflammatory protein VGF. However, most of these studies focused on BDNF signalling in the hippocampus [83,84,85,86,87]. It is widely reported that voluntary running results in an increase in the BDNF expression in the hippocampus, cerebral cortex, and other areas of the brain [88], as well as the protein level of BDNF in the sera of physically active humans [89,90,91]. In previous studies, it has been established that a long-term exposure to exercise reverses chronic stress-induced memory deficits via the BDNF induction [92], and molecular mechanisms regulating this plasticity may include the SIRT1/microRNA, CREB/BDNF, and AKT/GSK-3β signalling pathways [93]. Physical exercise and stress have been shown to modulate, in a different manner, the brain expression of BDNF transcripts by possibly different epigenetic mechanisms [94]. In particular, the BDNF transcripts expression induced by exercise was associated with increased levels of H3Ac, while no post-translational histone modifications induced by two-hour restraint stress were found (Table 3).

Rats subjected to six weeks of endurance training before acute stress developed a stress resilience and reflected reduced anxiety and reduced ASR (Acute Startle Response), which corresponded with the increased expression of BDNF, neuropeptide Y (NPY), and phospho-delta opioid receptor (DOR) signalling in the hippocampus. It might be suggested that an increase in the BDNF, NPY, and DOR enhances the synaptic plasticity, leading to stress resilience [95]. In a recent study, it was proposed that the BDNF/TRKB pathway is involved in the enhancement of resilience by voluntary exercise. Voluntary exercise promotes resilience to stress, through the exercise-regulated PGC1a/FNDC5 pathway known to induce hippocampal BDNF signalling. Exercise increases the levels of the transcriptional coactivator PGC1a, which in turn activates the expression of FNDC5. FNDC5 is a protein that is processed and secreted, and that induces BDNF/TRKB activation. The results obtained suggest that exercise may mediate stress resilience by engaging the BDNF/TRKB signalling pathway [96]. 

Wheel running both before and concurrently with CORT administration in rats prevented the depressive phenotype and reductions in neurogenesis that normally accompany a high-dose CORT administration [97]. In the hippocampi of these animals, an increase in the hippocampal levels of synaptophysin and PSD-95 protein levels, but no changes in the levels of BDNF or IGF-1, was observed. The results presented above suggest that exercise can promote stress resilience by enhancing hippocampal neurogenesis and increasing the synaptic protein levels, thereby reducing the deleterious effects of stress. Furthermore, in the study by Munive et al. (2015), treadmill running for two weeks enhanced the resilience to stress in female mice only. Authors have postulated that exercise modulates mood in both sexes acting through different mechanisms addressing distinct components of mood. Additionally, exercise increased the hippocampal levels of IGF-1 only in cycling females. IGF-1 was proposed as a factor probably involved in the mechanism underlying the observed differences in sex incidence of mood disorders [98,99]. The BDNF is a known cAMP target gene involved in stress-related disorders. Adenylyl cyclase (ADCY) and cAMP regulate synaptic plasticity, which may be involved in coping with stress [100]. In a recent study by Yang et al. (2020), the overexpression of *Adcy1* prevented the two-hour restraint-induced down-regulation of BDNF and NPY. This overexpression caused the concurrent elevation of the basal CORT level and behavioural resilience to stress, which suggests that *Adcy1* may play a role in stress resilience [101]. Finally, a very recent study provided results that strongly support previous reports that prior exercise can prevent the harmful effects of exposure to stress through, among other routes, the BDNF modulation. In a rat model of PTSD, four weeks of treadmill exercise protocol ameliorated stress-induced enhanced anxiety levels, serum CORT, the BDNF protein level in the hippocampus and serum, and apoptosis markers (Bax, Bcl-2, and Caspase 3) as compared to the sedentary rats [102]. 

**Table 3 ijms-23-13348-t003:** Summary of animal research focusing on neurotrophic factors.

No.	Strain	Sex	Age	Physical Activity	Stress Model	Main Outcome	Reference
1	C57BL/6	male	8 weeks	voluntary wheel running	acute restraint stress	elevated expression of BDNF transcripts in the hippocampus	[94]
2	C57BL/6	male and female	8–9 weeks	treadmill	forced swim and tail suspension	increased IGF-1 level in the hippocampus	[99]
3	C57BL/6J	male	2.5–3.5 months	voluntary wheel running	physical restraint	increased Adcy1 expression in the forebrain	[101]
4	Sprague–Dawley	male	7–8 weeks	treadmill	predator scent stress	increased expression of BDNF, NPY, and DOR signalling in the hippocampus	[95]
5	Sprague–Dawley	male	8 weeks	voluntary wheel running	depression	enhanced hippocampal neurogenesis; increased synaptic protein levels	[97]
6	Wistar	male and female	7–8 weeks	treadmill	posttraumatic stress disorder	increased BDNF level and apoptotic markers in the hippocampus	[102]

Adcy1—type 1 adenylyl cyclase; BDNF—brain-derived neurotrophic factor; DOR—delta opioid receptor; IGF-1—insulin-like growth factor 1; NPY—neuropeptide Y.

### 4.4. Hypothalamic–Pituitary–Adrenal Axis Activity

Physical exercise alters the action of the HPA axis [103], and the response of the HPA axis to both acute and chronic stress is changed in exercised rodents [104,105]. The HPA activity may differ depending on the exercise model (voluntary versus forced) and the duration (short-term or long-term) [104,106,107]. The results from the experimental studies are inconsistent, depending on the stress stimuli (acute vs. chronic), when HPA axis-mediated responses are studied. While some reports identify behavioural and biochemical changes in response to acute stress with prior exercise, limited studies are available regarding the effects of long-term stress on the HPA responses in exercise rodents (Table 4). In one study utilizing the rat model of PTSD (one week of social defeat), which involves aggressive encounters between the resident and intruder, prior treadmill exercise was found to be protective against depressive and anxiety-like behaviour. Two weeks of moderate treadmill exercise provided a resilience to following stressful stimuli by modulating the CORT levels [108]. Campenau et al. (2010) presented data showing that six weeks of voluntary wheel running in rats decreased the HPA response (assessed by plasma levels of ACTH and CORT) to low-intensity stressors such as an exposure to a novel environment accompanied by a central mechanism. In particular, exercise resulted in a lower *c-fos* mRNA expression following stress in the paraventricular nucleus of the hypothalamus [104]. Relatively few studies have reported that exercise enhances the HPA axis response to subsequent acute restraint stress. Four weeks of voluntary wheel running reduced anxiety-like behaviour, which was associated with a more rapid CORT response and CORT decay to acute restraint stress, increased adrenal gland size, and elevated sensitivity to ACTH [109]. These observations suggest that exercise may promote stress resilience partly by the different responses of HPA to a stressor, thus affecting the overall exposure to the possible negative effects of a more sustained activation of the HPA axis. Similarly, voluntary running for three weeks before the two-hour immobilization of stress resulted in increased CORT levels both immediately after and ten hours after the cessation of the stressor. Authors have suggested that CORT mediates stress-induced decreases in the BDNF but is not a primary mediator for exercise-related increases in the BDNF levels [5]. Further, treadmill exercise for two weeks modulates the acute stress-induced release of ACTH and has a noradrenergic effect in the brain areas of female rats. It has been suggested that two weeks of exercise protects against the stress-induced NE decrease in the amygdala, hippocampus, and LC of rats [110]. Another positive impact of exercise in regard to stress resilience was connected with an increase in the hippocampal expression of *Nr3c1*, which was correlated with a down-regulation of miR-124, an epigenetic regulator of *Nr3c1* [111].

Sasse et al. (2013) showed changes in the corticotropin-releasing factor (CRF) and BDNF systems in several brain regions that were associated with a habituation to repeated noise stress in prior exercised rats [112]. On the other hand, rats exposed to chronic isolation plus swimming exercise were shown to have fewer deleterious effects induced by social isolation stress on the limbic-HPA axis, as assessed by the brain levels of GR and Hsp70 [113]. Furthermore, heat shock proteins (HSPs), induced by a variety of stressors, have been implicated in stress resilience. Prior voluntary running facilitates the inescapable tail-shock stress-induced HSP72 induction in the rat brain regions [114]. A greater and faster increase in HSP72 after stress in the exercised rats may contribute to exercise-induced stress resilience at the cellular level. 

**Table 4 ijms-23-13348-t004:** Summary of animal research focusing on HPA (hypothalamic-pituitary-adrenal) axis activity.

No.	Strain	Sex	Age	Physical Activity	Stress Model	Main Outcome	Reference
1	C57BL/6	male	2 months	voluntary wheel running	restraint stress	modulation of serum CORT level, increased BDNF protein level in the hippocampus	[5]
2	C57BL/6J	male	6 weeks	voluntary wheel running	restraint stress	reduced CORT response duration; increased adrenal sensitivity	[109]
3	C57BL/6	N/A	8 weeks	voluntary wheel running	social stress (housing conditions)	increased hippocampal expression of glucocorticoid receptor (Nr3c1), correlated with a down-regulation of miR-124	[111]
4	Wistar and Sprague–Dawley	male	N/A	treadmill	social stress	decreased serum CORT levels induced by stress; alleviated behavioural consequences of stress	[108]
5	Sprague–Dawley	male	2 months	voluntary wheel running	low-intensity stressors	reduced HPA responses to stressors	[104]
6	Sprague–Dawley	female	40 days	treadmill	restraint stress	augmented NE decrease in the amygdala, hippocampus, and LC after which modulate the regulation of ACTH	[110]
7	Wistar	male	2 months	voluntary wheel running	repeated noise stress	regulation of CRF and BDNF expression in several brain regions	[112]
8	Wistar	male	3 months	swimming	chronic social isolation or immobilization or cold (4 °C)	enhanced GR and Hsp70 levels in the hippocampus and brain cortex	[113]
9	Fischer F344	male	8–9 weeks	voluntary wheel running	tail-shock stress	facilitated induction of HSP7 in the brain	[114]

ACTH—adrenocorticotrophin; CORT—corticosterone; CRF—corticotropin-releasing factor; GR—intracellular glucocorticoid receptor; Hsp70—heat shock protein 70; LC—locus coeruleus; NE—norepinephrine.

### 4.5. Oxidative Stress 

Exercise protects against chronic stress-induced deleterious effects, namely an increase in oxidative stress markers. The induction of oxidative stress is associated with anxiety-like behaviour in rats. Namely, acute sleep deprivation increases the oxidative stress in the cortex, hippocampus, and amygdala, while a prior treadmill exercise prevents this increase [115] (Table 5). The authors of this study proposed that the brain expression of enzymes involved in the oxidative stress pathway, glyoxalase-1 (GLO1) and glutathione reductase-1 (GSR1), are associated with anxiety-like behaviours. Moreover, the reported prevention of anxiety-like behaviour observed in exercised rats might be mediated via oxidative mechanisms [116]. Another anti-oxidative potential of preconditioning exercise arose from the study by Gerecke et al. (2013). In this study, restraint stress caused the up-regulation of proapoptotic Bax protein in the mouse cortex, but not the hippocampus. Voluntary wheel running alone resulted in a decreased Bax level in comparison to stressed sedentary animals, and also Cox-2 expression in the hippocampus in relation to the control and stressed sedentary mice. Moreover, exercise protected against an increase in the number of hippocampal Iba1/Cox-2 positive cells [116]. Treadmill exercise introduced before chronic variable stress (CVS) improves aversive memory and oxidative status changes in the amygdala and hippocampus, as well as Na^+^, K^+^-ATPase activity in the hippocampus of rats. The results obtained suggest that physical activity applied in developmental life stages ameliorates the brain resilience to stress later in life [117].

**Table 5 ijms-23-13348-t005:** Summary of animal research focusing on oxidative stress.

No.	Strain	Sex	Age	Physical Activity	Stress Model	Main Outcome	Reference
1	C57/B16J	female	8 weeks	voluntary wheel running	restraint stress	Enhanced expression of proapoptotic Bax protein in the mouse cortex and microglia/macrophage expression of Cox-2 in the hippocampus	[116]
2	Wistar	male	45–48 days	treadmill	sleep deprivation	increased protein expression of GLO1 and GSR1 in the amygdala, cortex, and hippocampus; ameliorated anxiety-like behaviour	[115]
3	Wistar	male	3 weeks	treadmill	chronic stress restraint and forced swim stress	improved aversive memory and oxidative status changes in the amygdala and hippocampus, Na^+^, K^+^-ATPase activity in the hippocampus	[117]

GLO1—glyoxalase 1; GSR1—glutathione reductase 1; LC—locus coeruleus.

### 4.6. Galanin System in Locus Coeruleus 

Galanin, widely distributed in the brain, modulates stress, mood, cognition, food intake, nociception, and seizures [118]. This neuropeptide is synthesized in noradrenergic LC, which is a region activated by stress, and NE release plays a crucial role in the stress response [119]. Chronic clomipramine treatment resulted in an increase in galanin expression in the LC in a similar way to exercise [120]. The exercise and galanin treatment each prevented anxiety-like behaviour after stress [121]. Moreover, mice with a knockout for the galanin receptor (GalR2) showed anxiety- and depressive-like behaviours, suggesting GalR2 may be involved in the antidepressant-like effects of galanin and may be a valid drug target for depressive disorders [122]. Moreover, the activation of GalR2 mediates a variety of actions related to neurite dynamics and the mechanisms for protection against the dendritic spine atrophy induced by stress [123,124]. The maintenance of the dendritic spine in some stress-vulnerable brain regions has been suggested as a cellular mechanism for brain resilience [125,126]. In rats, three weeks of wheel running increased the galanin levels in the LC, and this was associated with behavioural resilience as a result of exercise [127,128]. Recently, Tillage et al. (2020) have demonstrated findings that support the important role of exercise and galanin in stress resilience (Table 6). These authors have shown that the overexpression of galanin in noradrenergic neurons in mice ameliorates stress-induced anxiety-like behaviour [129].

### 4.7. RNA Methylation 

The epitranscriptome refers to the dynamic and reversible chemical post-transcriptional modification of RNA. The most abundant internal RNA modification is the generation of N6-methyladenosine (m6A), which occurs in a region-specific manner within the brain, reviewed in [130]. In the CNS, the m6A signalling is involved in several physiological processes [131], neurodevelopment [132,133,134], cognitive functions [135], and stress response, reviewed in [136]. In mice, restraint stress leads to global methylation changes in the PFC and amygdala. Moreover, the deficiency of *Mettl3* and *Fto* indicates the role of m6A/m regulatory genes in stress resilience [137]. m6A-mRNA methylation in the VTA increases in response to CUMS in comparison to unstressed animals [138]. Yan et al. (2022) showed that treadmill exercise in RNA m6A homeostasis in the PFC plays a role by improving the resilience against CRS for one hour for 14 days [139]. RNA methylation in mPFC neurons depends on methyl donor S-adenosyl methionine (SAM) liver biosynthesis. CRS suppresses the cortical and circulating SAM levels, and physical activity potentiates the liver synthesis. The results indicate that *Map1a* (methionine adenosyltransferase 1A) hepatic expression is essential for this protective effect of physical activity against the CRS. In summary, treadmill exercise maintains epitranscriptome homeostasis, affecting neural activity and reducing anxiety-like behaviour via the liver–brain pathway as a mechanism of stress resilience [140].

**Table 6 ijms-23-13348-t006:** Summary of animal research focusing on galanin system and RNA methylation.

No.	Strain	Sex	Age	Physical Activity	Stress Model	Main Outcome	Reference
1	C57BL/6J	male and female	3–6 weeks	running wheels	foot shock	increased galanin expression in noradrenergic neurons	[129]
2	C57BL/6	male	5–7 weeks	treadmill	restraint stress	improved brain RNA m6A methylation	[139]
3	Sprague–Dawley	male	7 weeks	voluntary wheel running	foot shock	increased galanin levels in the LC	[127]

LC—locus coeruleus.

## 5. Limitation of the Study

It should be noted that there are some discrepancies in this review study compilation. First, there are various forms of physical exercise, such as running and swimming, and these differ in duration and intensity. Additionally, the motivation of animals to engage in physical activity (voluntary or forced) has an impact on their response to stress. Even though forced treadmill running induces a stress response, manifested as increased CORT levels and anxiety-like behaviour, the role of physical activity in stress resilience is undoubted. Second, in the same way as animal physical activity models, stress response depends on the type and duration of the introduced stimuli. Therefore, the changes induced by different stress models may also be contradictory. 

Based on the literature, it might be suggested that physical activity does affect stress resilience, but the initial phenotype (stress-resistant or more susceptible) has a great impact on the response to stress and this phenotype might influence the direction of changes induced by physical activity. In support of this, voluntary wheel running prior to chronic stress increased the proportion of resilient mice [66,96]. On the other hand, the initial exercise sensitivity of animals (related to a low and high adaptive response to exercise) should be taken into account since it is known that the acquired exercise capacity plays a role in regulating the response to stress [65]. Therefore, it appears quite straightforward that the increased physical activity of individuals with a stress-resistant phenotype may prevent them from developing psychopathology. More importantly, on the basis of reviewed publications, we may postulate that neurobiological mechanisms of exercise-induced brain resilience are diverse and complex and need to be further explored. However, we can summarize that physical activity may promote the development of a stress-resilient phenotype. The proposed mechanisms which involve how exercise can promote the development of the stress-resilient phenotype brain resilience are presented in Figure 2.

Finally, the studies mentioned were conducted on both mice and rats of different ages. Moreover, the underrepresentation of female animal research possibly contributes to a divergence in explaining brain resilience to stress among genders. Numerous studies may relate to the question of whether the behavioural paradigm measures acute stress reactivity or the continuation of the brain coping with stress. Anxiety- and depression-like behaviours are also taken into consideration as determinants of stress resilience in the above literature. Therefore, these limitations should be considered in future studies.

## 6. Conclusions and Future Directions

The ability to cope with stress is crucial for maintaining appropriate behavioural adaptation and for mental health. Factors that promote resilience to stress may be important for treatment strategies for CNS disorders such as anxiety and depression. As summarized above, a number of animal studies have shown that regular exercise can reduce or prevent stress-induced behavioural, biochemical, and molecular abnormalities. Exercise can significantly alter the CNS expression pattern of several genes and pathways strongly related to vulnerability to stress, and various molecules have been already identified. 

A full understanding of the neurobiological adaptations during exercise is critical in order to develop the rational and optimal treatment of many disorders and has the translational benefit of gaining an understanding of the contribution of physical exercise to stress resilience in the general population, but it may have a special importance in numerous psychiatric as well as neurological conditions. These include numerous mental [141,142], neurodegenerative, and other disorders [143], many of them related to aging [144]. Particular emphasis should be paid to the long-term effects of the COVID-19 pandemic. Maintaining good levels of physical activity during lockdown was a protective factor against developing stress-related symptoms in older people [145,146,147]. Some studies on human subjects and clinical observations confirm the importance of physical activity and exercise in attenuating neurobehavioral disturbances [148,149,150]. This deserves a separate analysis of the current status of scientific investigations. The role of the blood-brain barrier (BBB) in these processes should receive more attention in the future. The BBB is formed by tightly connected endothelial cells, which together with pericytes and astrocytes, regulate the exchange of nutrients and systemic factors between the brain and the periphery, reviewed in [151]. As shown in recent studies, the BBB and its neuroprotective role is altered in many neurological disorders and in response to stress and physical activity [152,153,154]. Molecules involved in stress-related disorders and in response to physical activity, such as glutamate, cAMP, glucocorticoids, oestrogen, and pro-inflammatory cytokines, can alter the expression of the prominent BBB tight junction protein claudin-5 either by direct or indirect mechanisms, resulting in a change in the BBB permeability with direct effects on the neuronal function, reviewed in [152]. Claudin-5 as a direct oestrogen target [155,156,157] is one of the proteins which mediates sex differences in neurological disorders such as depression and anxiety [158,159]. Additional molecular targets involved in the cellular responses to physical activity and different types of stress and stress resilience need to be identified.

## Figures and Tables

**Figure 1 ijms-23-13348-f001:**
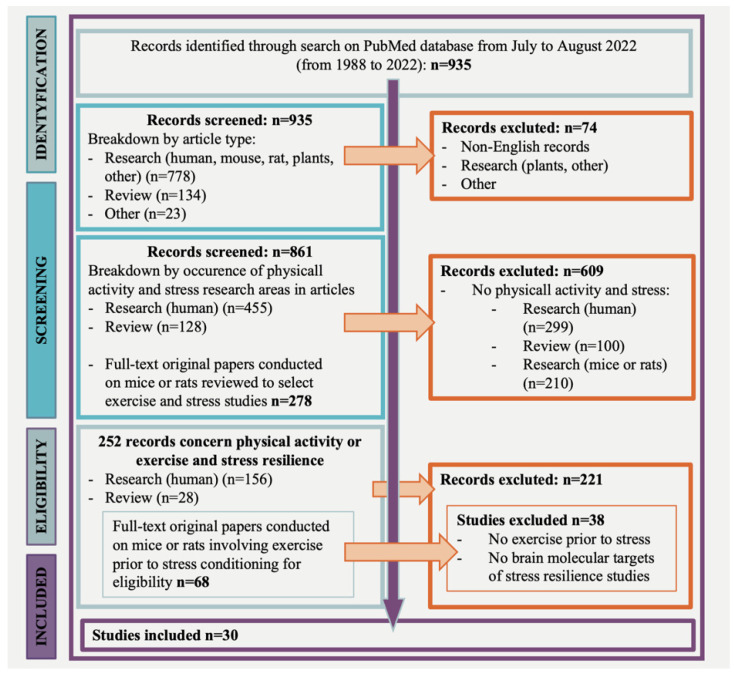
Flow chart of research strategy.

**Figure 2 ijms-23-13348-f002:**
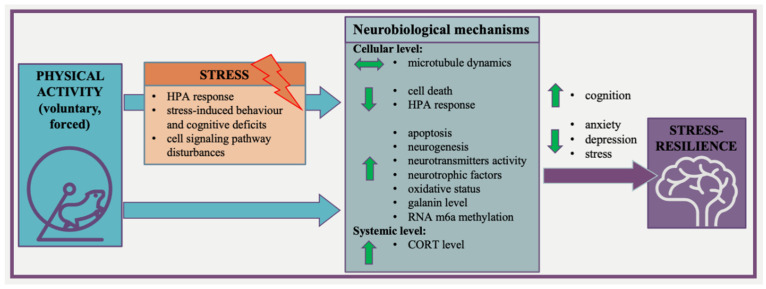
The proposed mechanisms involved how exercise can promote the development of the stress-resilient phenotype brain resilience.

## Data Availability

Not applicable.

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
