# Peer review of "Running from Stress: Neurobiological Mechanisms of Exercise-Induced Stress Resilience"

_ijms, 2022, doi:10.3390/ijms232113348_

Round 1

Reviewer 1 Report

The manuscript provides significant new interesting information and scientifically sounded concepts. The purpose to describe the actual problem of stress resilience is clearly stated. The information is of significant interest to the readers.

The data areclearly presented with (a) an introduction stating the objectives and background, (b) precisely described flowchart of literature search, (c) clearly described results with tables presented with maximum clarity, and (d) a relevant and concise bibliography (the references cited the most appropriate to support the manuscript).

The title is succinct, and the abstract is clearly summarizing the background, methodology, results, and significance of the study

The topic of the manuscript is appropriate for the Journal. The paper conforms to the Journal scope and manuscript preparation guidelines. Writing style and English usage are appropriated.

It will be interesting to recognize what authors think why treadmill running for two weeks did enhanced resilience to stress in female only, not in male mice? Additionally, exercise increased hippocampal levels of IGF-1 only in cycling females, how in different phases?

Author Response

We would like to thank the Reviewer for taking the time and effort necessary to provide all comments.  

Reviewer #1

Comments and Suggestions for Authors.

The manuscript provides significant new interesting information and scientifically sounded concepts. The purpose to describe the actual problem of stress resilience is clearly stated. The information is of significant interest to the readers.

The data are clearly presented with (a) an introduction stating the objectives and background, (b) precisely described flowchart of literature search, (c) clearly described results with tables presented with maximum clarity, and (d) a relevant and concise bibliography (the references cited the most appropriate to support the manuscript).

The title is succinct, and the abstract is clearly summarizing the background, methodology, results, and significance of the study

The topic of the manuscript is appropriate for the Journal. The paper conforms to the Journal scope and manuscript preparation guidelines. Writing style and English usage are appropriated. 

It will be interesting to recognize what authors think why treadmill running for two weeks did enhanced resilience to stress in female only, not in male mice? Additionally, exercise increased hippocampal levels of IGF-1 only in cycling females, how in different phases?

Response:

We would like to thank the Reviewer for taking the time and effort necessary to provide all comments.  

As the Reviewer noticed, in the study by Munive et al. (2015), the Authors showed that two weeks of treadmill running enhanced resilience to stress in female mice only which was associated with an increase of hippocampal levels of IGF-1 only in cycling females. Results from animal and clinical studies showed a sex bias in CNS disorders with a sex-specific range of occurrence. Puberty was proposed as the beginning of that difference, but hormonal changes aren’t the only reason. Regarding stress, epidemiological studies indicate that long-lasting exposure to stress has become a key lifestyle modification over the last decades. Chronic stress, even daily life-related stress of moderate intensity, is widely acknowledged as a predisposing or precipitating factor in neuropsychiatric diseases, especially in women. Data from animal research revealed that the physiological and behavioral responses to environmental factors are sex-dependent [Oyola MG et al.,  Stress. 2017;20 (5):476–94. doi:10.1080/10253890.2017.1369523; Vieira JO et al., Prog Neuropsychopharmacol Biol Psychiatry. 2018;81:426–37. doi:10.1016/j.pnpbp. 2017.08.014.]. For example, major depression affects about 20% of the population, with females, especially in reproductive years, outnumbering males by 2:1 (Albert PR. J Psychiatry Neurosci. 2015 Jul;40(4):219-21. doi: 10.1503/jpn.150205) This difference is at least partially a consequence of different responses of the stress-coping system of the female and male brain to adverse life experiences. Physical activity is known to boost mood and ease depression symptoms, but getting a lot of exercises is extra important for women. IGF-1 was proposed as a factor probably involved in the mechanism underlying observed differences in sex incidence of mood disorders [Grunbaum-Novak et al., Eur Neuropsychopharmacol. 2008;18(6):431-8. doi: 10.1016/j.euroneuro.2007.08.004.]. It is well known that estrogens and IGF-1 interact with each other, modulating mood [Nelson et al., Psychopharmacology (Berl). 2014; 231(5):899-907. doi: 10.1007/s00213-013-3310-7]. Circulating and brain levels of IGF-1 are sex- and age-depended [Ashpole et al., Geroscience. 2017 ;39(2):129-145. doi: 10.1007/s11357-017-9971-0]. We agreed with Munive et al. (2015), that exercise modulates mood in both sexes acting through different mechanisms addressing distinct components of mood. Authors postulated that sex differences in hippocampal IGF-I are established during the so-called “organizational period” taking place early during the sexual differentiation of the brain, whereas regulation by the exercise of the adult sex-dependent pattern is sensitive to ovarian steroids such as estradiol. 

We have now added some comments regarding the resilience to stress in females and IGF-1 in the revised manuscript. All changes are highlighted in yellow. 

Reviewer 2 Report

This is a well-documented and literature review with extensive references. The text and Figure 2 present a clear rendition as to the criteria by which the final 32 studies were derived. The conclusions and limitations are consistent with the studies reviewed.

In reading the abstract and the Introduction, particularly lines 99-101, I do not clearly discern the purpose of the study as well as the role of the 32 studies. In the reading the Results section it is not clear to me how/why the categories presented are derived from the literature. Perhaps a concluding paragraph in the Introduction and the Abstract is appropriate.

One may initially to conclude that the focus of the paper is the 32 studies by looking at Figure 1. Upon examination a substantial number of studies are cited beyond the 32. In the presentation these studies appear to be minimized relative to other studies cited. My conclusion is the study is not a literature of 32 studies, nor are they systematically analyzed. I would expect a more nuanced analysis of the 32 studies using meta-analysis or a formal content analysis.

Author Response

Reviewer #2

Comments and Suggestions for Authors 

This is a well-documented and literature review with extensive references. The text and Figure 2 present a clear rendition as to the criteria by which the final 32 studies were derived. The conclusions and limitations are consistent with the studies reviewed.

In reading the abstract and the Introduction, particularly lines 99-101, I do not clearly discern the purpose of the study as well as the role of the 32 studies. In the reading the Results section it is not clear to me how/why the categories presented are derived from the literature. Perhaps a concluding paragraph in the Introduction and the Abstract is appropriate.

One may initially to conclude that the focus of the paper is the 32 studies by looking at Figure 1. Upon examination a substantial number of studies are cited beyond the 32. In the presentation these studies appear to be minimized relative to other studies cited. My conclusion is the study is not a literature of 32 studies, nor are they systematically analyzed. I would expect a more nuanced analysis of the 32 studies using meta-analysis or a formal content analysis.

Response:

Thank you for your valuable suggestions. 

The Abstract and Introduction sections were revised with a special emphasis on the purpose of the study. We agreed that the description of the Results section could mislead the Reader. The Result section was improved and now is consistent with the revised Flow chart. All changes are highlighted in yellow. 

In our review, we tried to discuss the findings only from animal studies of the molecular and psychobiological mechanisms underlying the preventive effect of exercise on stress. A Pubmed database search allowed us to choose 32 stexercise and stress resilience studiesIn the revised manuscript we have excluded 2 studies and we have presented 30 original studies. These 30 works were the scaffolding for discussion, however as the Reviewer has noticed we cited studies beyond the 30. We have referenced additional studies to discuss more complex possible mechanisms involved in exercise-induced stress resilience. We choose the categories/subparagraphs based on our search of the literature and on the categories that are known to be involved in exercise-induced neuroprotection. In the revised version of the manuscript, we have added new tables in the Discussion section, a supplementary table, and a scheme with the proposed mechanism involved in how exercise can promote brain resilience. 

Reviewer 3 Report

Dear authors, after significant revision, the manuscript may be published.

Author Response

We would like to thank the Reviewer for taking the time and effort necessary to provide all comments and suggestions.  

Reviewer #3

  1. In the Introduction to the manuscript, the authors indicated that after exposure to stress, only a portion of the people developed various pathologies. In contrast, the other part developed adaptive physiological and psychological reactions (resistance to stress). Such differences in individual responses to stress (resistance/susceptibility) depend on a combination of genetic and non-genetic factors that interact with each other in complex ways to form the initial phenotypes (stress-resistant and stress-vulnerable). It should be noted that stress-resistant individuals are characterized by high sympathetic system activity, low parasympathetic system activity (for example, high heart rates), as well as low HGAS reactivity (low production of glucocorticoids in response to stress) [DOI: 10.1016/s0149-7634(99)00026 -3, DOI: 10.1016/j.yfrne.2010.04.001], high resistance to mental illness [DOI: 10.1016 / j.yhbeh.2014.08.004] and high motor activity (active exploratory reactions, aggression, etc.). A study by Jing Zhang et al. (2021) showed that mice with high baseline physical activity (high voluntary activity in "running wheels") demonstrated resistance to chronic social stress compared to animals neglecting exercise [DOI: 10.1016/j.euroneuro. 2021.02.011]. 

In connection with this, several questions arise: A) To what extent is physical activity a 'therapy' for stress? Probably, the data obtained in various studies indicate only the predominance of animals with a stress-resistant phenotype in the experimental samples (especially since linear animals were used most often in the experiments). In my opinion, the article should be supplemented with data on the effect of physical activity on individuals, divided into stress resistant and stress susceptible (add a section). 

  1. C) The authors, when analyzing the literature (in Table 1), point out the differences in the design of experiments (by duration, intensity, and type of stress; by duration, intensity, and type of physical activity, by motivation for physical activity - arbitrary or forced). At the same time, most of the works used in this review describe experiments with voluntary physical exercises. Please answer the following question: Does physical activity affect stress resistance? Or does the initial phenotype determine the physical activity of the individual and, as a result, prevent them from developing psychopathology? C) Similar remarks (indicated in Subsection "B") apply to the timing of physical training (before or after stress).

Response:  

We would like to thank the Reviewer for taking the time and effort necessary to provide all comments and suggestions.  

Regarding 1) We have added the Reviewer’s suggestion to the Introduction and Discussion sections. All changes are highlighted in yellow.

Regarding subsection A) In recent years several experimental and clinical studies, have shown data supporting the role of physical activity in reversing or minimizing stress-induced changes. However, in our review, we made an effort to discuss the results of experimental studies in which activity precedes exposure to stress. In this context, initial exercise sensitivity plays a crucial role in subsequent stress exposure and may serve as a kind of “preconditioning”. Vanderheyden et al., 2020 showed that rats selectively bred for high adaptation for exercise training have higher molecular and structural brain adaptations in response to stress. Therefore, it is difficult to address the Reviewer’s question of whether these studies only show that most of the animals were resistant to stress. We agree with the Reviewer that such a division based on stress-resistant and vulnerable to stress phenotypes should be a standard procedure in animal experiments addressing the impact of stress, especially in the context of CNS. According to the Reviewer’s suggestion, we addressed this issue In the revised version of the manuscript and added a new paragraph in the Limitation of the study. 

Regarding subsection B) 

 A Pubmed database search allowed us to choose 32 exercise and stress resilience studies. These 32 works were the scaffolding for discussion, however as the Reviewer has noticed we cited studies beyond the 32. We have referenced additional studies to discuss more complex possible mechanisms involved in exercise-induced stress resilience. To sum up, the articles that were revealed after advanced Pubmed database searching. In the revised manuscript we have  listed 30 articles (2 articles were excluded) in a few tables. Voluntary wheel running was used in 18 studies, which is a result of the above-mentioned literature search.

Based on the literature it might be concluded that physical activity does affect stress resilience, but the initial phenotype has a great impact on response to stress and this phenotype might influence the direction of changes induced by physical activity. In our opinion, increased physical activity of individuals with a stress-resistance phenotype may prevent them from developing psychopathology. We definitely agree that individual responses to stress depend on various factors (genetic, and environmental), but the complexity of the pathogenesis of stress-resistance or - susceptibility phenotypes are not always possible to be modeled in animal studies.

Regarding C)

In Table 1 we have listed studies utilizing the exercise paradigm (voluntary wheel running, forced: treadmill, or swimming exercise) before stress. Studies that did not address voluntary physical activity or exercise before or simultaneously with the stress paradigm were excluded. Now we have changed the presentation of the Pubmed search in the Discussion. New Tables were added to the revised manuscript. This issue is also addressed in the Limitation of the study section, and we also added a supplementary table S1. 

  1. It is advisable to supplement each section of the table with data from the literature from studies in which animals were forced to exercise, as well as data on the effect of physical activity on individuals, divided into resistant to stress and resistant to stress. In addition, in order to improve the perception of information, it is necessary to supplement the existing table with the main research results or add a new table. 

Response: 

According to the Reviewer’s suggestion, we have added new tables to each subsection of the Discussion, and an additional supplementary table S1. 

  1. In conclusion, the authors point out the possibility of using physical exercises as a therapy in people with various mental and neurological conditions. Please provide examples.
  2. The authors write: "Some studies on human subjects and clinical observations confirm the importance of physical activity and exercise in attenuating neurobehavioral disturbances" (lines 445-447). Links must be added. 

Response: The conclusion section has been improved. Please see the revised version of the manuscript. 

  1. After the revision of the manuscript, its title will need to be changed.

Response: We have changed the title as suggested:” Running from stress: neurobiological mechanisms of exercise-induced stress resilience.”

  1. If the authors of the article declare the mechanisms of the protective effect of physical exercises in psychopathologies caused by stress loads, then in conclusion, based on the analysis of literary data, the authors should propose a pathogenetic scheme explaining the protective effect of physical exercises.

Response:

We have added a scheme (please see Figure 2) with the proposed mechanism involved in how exercise can promote brain resilience. 

Round 2

Reviewer 2 Report

The authors are commended for their thoughtful revision of the manuscript.  Well done.

Reviewer 3 Report

Dear authors, thank you for the revision.